# Identification of an Acidic Amino Acid Permease Involved in d-Aspartate Uptake in the Yeast *Cryptococcus humicola*

**DOI:** 10.3390/microorganisms9010192

**Published:** 2021-01-18

**Authors:** Daiki Imanishi, Yoshio Kera, Shouji Takahashi

**Affiliations:** Department of Bioengineering, Nagaoka University of Technology, Nagaoka, Niigata 940-2188, Japan; s175037@stn.nagaokaut.ac.jp (D.I.); yoshkera@nagaokaut.ac.jp (Y.K.)

**Keywords:** d-aspartate oxidase, amino acid permease, *Cryptococcus humicola*, d-aspartate, gene expression

## Abstract

d-aspartate oxidase (DDO) catalyzes the oxidative deamination of acidic d-amino acids, and its production is induced by d-Asp in several eukaryotes. The yeast *Cryptococcus humicola* strain UJ1 produces large amounts of DDO (ChDDO) only in the presence of d-Asp. In this study, we analyzed the relationship between d-Asp uptake by an amino acid permease (Aap) and the inducible expression of ChDDO. We identified two acidic Aap homologs, named “ChAap4 and ChAap5,” in the yeast genome sequence. *ChAAP4* deletion resulted in partial growth defects on d-Asp as well as l-Asp, l-Glu, and l-Phe at pH 7, whereas *ChAAP5* deletion caused partial growth defects on l-Phe and l-Lys, suggesting that ChAap4 might participate in d-Asp uptake as an acidic Aap. Interestingly, the growth of the *Chaap4* strain on d- or l-Asp was completely abolished at pH 10, suggesting that ChAap4 is the only Aap responsible for d- and l-Asp uptake under high alkaline conditions. In addition, *ChAAP4* deletion significantly decreased the induction of DDO activity and *ChDDO* transcription in the presence of d-Asp. This study revealed that d-Asp uptake by ChAap4 might be involved in the induction of *ChDDO* expression by d-Asp.

## 1. Introduction

d-aspartate oxidase (DDO or DASPO, EC 1.4.3.1) is a peroxisomal enzyme that catalyzes the oxidative deamination of acidic d-amino acids to produce their corresponding α-keto acids and ammonia. DDO has been found in various eukaryotic organisms, ranging from fungi to mammals, but it is not present in prokaryotic organisms [1,2,3,4,5]. To date, *DDO* genes have been cloned from various eukaryotic organisms including human and mouse, and their enzymatic properties have been extensively studied [6]. DDO of the yeast *C. humicola* strain UJ1 (ChDDO) is the first DDO cloned from a microorganism and has high substrate specificity and high activity for d-Asp [5,7]. In fungi, DDO participates in the assimilation and detoxification of acidic d-amino acids [8]. In animals, the enzyme is considered to regulate d-Asp levels in relation to some important physiological functions, such as hormone secretion and neurotransmission, and suggested to associate with the psychiatric disorder schizophrenia in human [9,10,11]. Therefore, it might be important to reveal the environmental and protein factors that can affect the expression of *DDO* gene.

Yeast cells can grow on various l- and d-amino acids as carbon, nitrogen, or both sources [8,12,13]. Amino acids are intracellularly catabolized by various enzymes, such as oxidase, deaminase, transaminase, to yield nitrogen in the form of ammonium, glutamate, or glutamine as nitrogen donors in biosynthetic reactions [13]. On the other hand, the carbonaceous products from the catabolic reactions enter central metabolic pathways including tricarboxylic acid cycle and gluconeogenesis pathway to be used as carbon sources in biosynthetic reactions or energy source [13].

*Cryptococcus humicola* strain UJ1 (recently reclassified as *Vanrija humicola* strain UJ1) is a basidiomycetous yeast that can utilize d-Asp as a sole source of carbon, nitrogen, or both and expresses significant levels of ChDDO only in the presence of d-Asp in culture medium [5,8]. This inducible expression of ChDDO in response to d-Asp is regulated at the transcriptional level [8]. This acidic d-amino acid-dependent induction of DDO activity has also been reported in other organisms. For example, DDO activity in the fungus *Fusarium sacchari* var. *elongatum* strain Y-105 increases when it is grown on d-Asp or d-Glu as the sole source of carbon and nitrogen [14]. DDO activity in the yeast *Candida boidinii* strain 2201 increases when it is grown on d-Glu as the sole source of nitrogen [15]. The oral or intraperitoneal administration of d-Asp to mice increases DDO activity in the liver and kidneys [16]. The oral administration of d-Asp to pregnant rats increases DDO activity in the livers and kidneys of newborn rats [9]. These findings suggest that the inducible expression of DDO in the presence of acidic d-amino acids might be widely distributed in eukaryotic organisms. However, the induction mechanism remains to be elucidated.

In *C. humicola* UJ1, the induction of ChDDO expression by d-Asp is significantly repressed by the copresence of l-Asp in media [5,8]. This repression is considered to be caused by the decreased uptake of d-Asp via competitive inhibition by l-Asp. In fungi, amino acids are imported into cells by amino acid permeases (Aaps) located in the plasma membrane. Most yeast Aaps belong to the yeast amino acid transporter (YAT) family, a group of proteins in the amino acid-polyamine-organocation (APC) superfamily [17]. Aap homologs of the APC superfamily are found in fungi as well as animals, plants, and bacteria [18]. The typical YAT possesses 12 transmembrane regions (TMs) and functions as a symporter through proton-driven secondary active transport [19,20]. In the yeast *Saccharomyces cerevisiae*, 19 Aaps have been identified [17]. The yeast Aaps have different substrate specificities. Most yeast Aaps have high affinity and specificity for specific amino acids, whereas the general Aap Gap1p is involved in the uptake of all naturally occurring l-amino acids as well as some d-amino acids and amino acids analogs [20,21,22,23]. In *S. cerevisiae*, acidic l-amino acids are imported by the dicarboxylic amino acid transporter Dip5p, as well as Gap1p [24]. Dip5p mediates the high affinity and transport-capacity transport of l-Asp and l-Glu [24,25]. The Aaps specific for dicaboxylic amino acids have been also identified and characterized in *Aspergillus nidulans* (AgtA) and *Penicillium chrysogenum* (PcDip5) [26,27]. However, it is unclear whether dicarboxylic Aaps can also mediate the uptake of acidic d-amino acids and whether they are involved in the induction of *DDO* expression by d-Asp in yeast. In addition, the reason why multiple Aaps contribute to acidic amino acid uptake is unclear.

In this study, as the first step to elucidate the mechanism of ChDDO induction by d-Asp in *C. humicola* UJ1, we investigated the relationship between d-Asp uptake by a dicarboxylic Aap and the d-Asp–induced expression of ChDDO. To reveal the relationship, we identified an acidic Aap, ChAap4, and analyzed the expression of ChDDO in a *ChAAP4*-disrupted strain. Our results revealed that ChAap4 might participate in d-Asp uptake into cells and, consequently, indirectly the inducible expression of ChDDO, and the protein is essential for growth on d-Asp and l-Asp at a high alkaline pH, illustrating the involvement of d-Asp uptake by acidic Aap in the inducible expression of *ChDDO* in the presence of d-Asp.

## 2. Materials and Methods

### 2.1. Materials

Yeast nitrogen base without amino acids (YNB), yeast carbon base (YCB), and YNB without amino acids and ammonium sulfate (YNB w/o AA and AS) were obtained from Difco (Detroit, MI, USA). d-Asp was a generous gift from Mitsubishi Tanabe Pharmaceutical (Osaka, Japan). Other amino acids were procured from Nacalai Tesque (Kyoto, Japan). All other chemicals were purchased from Wako Pure Chemical Industries (Osaka, Japan), Nacalai Tesque, or Sigma-Aldrich (St Louis, MO, USA). DNA polymerase was purchased from Takara Bio (Shiga, Japan). PCR primers (Appendix A) were synthesized by Eurofins Genomics (Tokyo, Japan).

### 2.2. Strains, Media, and Growth Conditions

*C. humicola* strain UJ1 [5] was used as the wild-type strain. *C. humicola* strain UM3 [28], a *ura3* mutant derived from strain UJ1, was used as the host for gene disruption experiments. The yeast cells were routinely grown in YPD medium (1% (*w/v*) yeast extract, 2% (*w/v*) peptone, 2% (*w/v*) glucose), or SD medium (0.67% (*w/v*) YNB, 2% (*w/v*) glucose). Wherever required, 20 μg/mL uracil was added to SD medium. The growth test of *ChAAP*-disrupted strains (*Chaap*) was performed as follows. Cells were precultured in SD medium at 30 °C for 16 h. The preculture was inoculated into synthetic medium containing 1.17% (*w/v*) YCB with 10 mM of each nitrogen source at a cell density of OD_600_ = 0.005. The initial pH was adjusted to 7.0. For solid medium, 2% (*w/v*) agar was added to the culture medium. *Escherichia coli* strain DH5α was employed as a host for propagating plasmids. *E. coli* cells were grown at 37 °C in LB medium (1% (*w/v*) tryptone, 0.5% (*w/v*) yeast extract, 0.5% (*w/v*) NaCl) supplemented with 100 μg/mL ampicillin.

### 2.3. Identification of Aap Homologs

The putative Aaps of strain UJ1 were identified in the yeast draft genome sequence data [29] via a BlastP search [30] using the amino acid sequences of *C. neoformans* Aaps 1–8 [31]. A phylogenetic tree based on the amino acid sequences of Aaps was constructed by the neighbor-joining method using Molecular Evolutionary Genetics Analysis version 7 [32].

### 2.4. DNA and RNA Preparation

The total DNA of the yeast was prepared as described previously [28]. To prepare total RNA, cells were washed twice with ice-cold water, resuspended in ice-cold water, and then transferred to a 2 mL screw tube containing an equal volume of φ0.45–0.5 mm zirconia beads (BioSpec Products, Inc., Bartlesville, OK, USA). After centrifugation at 12,000× *g* for 2 min at 4 °C, the supernatant was removed, the cells were lyophilized using a freeze dryer system (DRC-1100 and FDU-2100, EYELA, Tokyo, Japan), and stored at −80 °C until use. The tube containing the lyophilized cells was shaken vigorously for 5 min using a vortex mixer. Total RNA was extracted and purified from the disrupted cells using a Direct-zol RNA Miniprep Kit (ZYMO Research, Irvine, CA, USA) according to the manufacturer’s instructions.

### 2.5. Disruption of ChAAP Genes

An approximately 2.0 kbp *ChURA3* fragment was amplified using Tks Gflex DNA polymerase (Takara Bio) with the primer set URA3F/URA3R (Appendix A) and the yeast vector pICUG as a template [28]. Approximately, 1.0 kb of the 5′- and 3′-regions of *ChAAP4* and *ChAAP5* was amplified using the following primer sets: AAP4UF/AAP4UR for the *ChAAP4* 5′-region, AAP4DF/AAP4DR for the *ChAAP4* 3′-region, AAP5UF/AAP5UR for the *ChAAP5* 5′-region, and AAP5DF/AAP5DR for the *ChAAP5* 3′-region (Appendix A). Yeast genome DNA was employed as a template. The amplified *ChURA3* and 5′- and 3′-fragments of each gene were mixed, and the fused DNA fragments were obtained via overlap extension PCR using the 5′- and 3′-terminal primers for each gene. The resulting disruption cassette for each *ChAAP* gene (approximately, 5.0 kbp) was introduced into the yeast cells via electroporation using a MicroPulser Electroporator (Bio-Rad, Hercules, CA, USA) as described previously [28]. Transformants were selected for their ability to grow on SD medium without uracil at 30 °C for 3 days. The disruption of *ChAAP4* and *ChAAP*5 was confirmed by PCR using the primer sets 4UFwd1/4URev1 and 4DFwd2/4DRev2 for *Chaap4* and 5UFwd2/5URev2 and 5DFwd3/5DRev3 for *Chaap5* (Appendix A). In these gene disruption mutants, *ChURA3* was inserted, in the opposite direction, into the genomic *ChAAP4* between 117 and 309 bp and *ChAAP5* between 716 and 1368 bp (Figure 1).

### 2.6. ChDDO Induction Experiment

Cells were precultured in SD medium at 30°C for 16 h. The preculture was inoculated into fresh SD medium at a final cell density of OD_600_ = 0.05 and further grown at 30 °C for 16 h with shaking at 166 rpm. The cells were collected by centrifugation and washed twice with ice-cold water, and 50 OD_600_ units of cells were resuspended in synthetic medium (YNB w/o AA and AS) or containing 60 mM d-Asp for each pH condition. After incubation at 30 °C for 3 h with shaking, the cells were harvested by centrifugation and washed twice with ice-cold water.

### 2.7. DDO Assay

DDO activity was measured as described by Takahashi et al. [8]. Briefly, cells were washed twice with ice-cold lysis buffer (50 mM potassium phosphate, pH 8.0, 2 mM EDTA), resuspended in the same buffer, and then transferred to a 2 mL screw tube containing an equal volume of φ0.45–0.5 mm glass beads. The tube was shaken vigorously for 1 min using a Mini Bead Beater-8 (BioSpec Products), followed by cooling on ice for 3 min. This procedure was repeated eight times. The extract was clarified via centrifugation at 20,000× *g* for 30 min at 4 °C, and the supernatant was used as the crude cell extract for the enzyme assay. The enzymatic activity was determined spectrophotometrically via a horseradish peroxidase (HRP)-coupled reaction with phenol and 4-aminoantipyrine (4-AA) [33]. The reaction mixture contained 20 mM d-Asp, 20 µM FAD, 2 mM phenol, 1.5 mM 4-AA, and 2.5 U/mL HRP (Sigma-Aldrich) in 50 mM potassium phosphate buffer (pH 8.0). The enzymatic activity was determined at 505 nm using a molar extinction coefficient for a generated quinone imine of 6580 M^−1^ cm^−1^.

### 2.8. Quantitative Real-Time RT-PCR (qRT-PCR)

Total RNA was isolated from cells grown in synthetic medium (YNB w/o AA and AS) containing 60 mM d-Asp. qRT-PCR of *ChDDO* transcripts was performed using RNA-direct SYBR Green Realtime PCR Master Mix (Toyobo, Osaka, Japan) with the primer set RTChDDOF2/RTChDDOR2 (Appendix A) in a StepOne real-time PCR system (Applied Biosystems, Foster City, CA, USA). Transcription of *TAF10* was determined as a normalizing gene using the primer set RTChTAF10F/RTChTAF10R, and the relative transcriptional levels of *ChDDO* were calculated against the normalizing gene using the 2^−ΔΔCt^ method [34].

### 2.9. Sequence Analyses and Structural Modeling

An amino acid sequence alignment was created using T-coffee (http://tcoffee.crg.cat/). Sequence comparisons were performed using EMBOSS Needle Pairwise Sequence Alignment (https://www.ebi.ac.uk/Tools/psa/emboss_needle/). Three-dimensional structure models were generated using I-TASSER (http://zhanglab.ccmb.med.umich.edu/) and visualized using PyMOL ver. 2.3 (http://www.pymol.org/).

## 3. Results

### 3.1. Identification of Acidic Aap Homologs of C. humicola Strain UJ1

To identify the Aap homologs of *C. humicola* strain UJ1, we searched for homologs in the draft genome sequence using the amino acid sequences of Aaps 1–8 of *C. neoformans* [31]. A BLASTP search identified 31 Aap homologs of strain UJ1 with significant similarities (E-value < 1.0 × 10^−50^ and amino acid sequence identity > 25%) to *C. neoformans* Aaps. It has been reported that the YATs responsible for the transport of similar amino acids (proline, acidic amino acids, basic amino acids, branched-chain amino acids, aromatic amino acids, and various amino acids) are classified into the same clade of a phylogenetic tree [17]. To predict which of the 31 Aap homologs of strain UJ1 is the acidic Aap, we constructed a phylogenetic tree using their amino acid sequences and those of known YATs (Figure 2). The phylogenetic tree illustrated that two Aap homologs, named “ChAap4 (g920) and ChAap5 (g5887),” belong to the same clade as acidic YATs (CnAap4, CnAap5, Dip5p, and AgtA).

ChAap4 and ChAap5 contained 573 and 540 amino acid residues, respectively, and they exhibited the highest amino acid sequence identities with CnAap4 (69.2% and 63.5%, respectively). In addition, ChAap4 displayed 41.5%, 41.7%, and 43.9% amino acid identities with Dip5p, PcDip5, and AgtA, respectively, and ChAap5 exhibited 40.1%, 38.9%, and 41.2% amino acid identities, respectively, with those same enzymes. Both *C. humicola* acidic Aap homologs were suggested to have 12 TM regions, and their N-terminal and C-terminal regions were predicted to be located in the cytoplasm, matching the typical topologies of YATs (Appendix A). In addition, the positions of the TM regions were mostly consistent with those of the *E. coli* arginine transporter AdiC. The GXG (where X is any amino acid) and (F/Y)(S/A/T)(F/Y)XGXE motifs were found in TM1 and TM6, respectively (Figure 3), and these motifs are highly conserved in Aaps and involved in the interactions with the α-carboxy and α-amino groups of amino acid substrates, respectively [17,35]. These findings suggested that ChAap4 and ChAap5 might function as acidic Aaps in strain UJ1. However, some amino acid residues conserved in dicarboxylic Aaps (AgtA, Dig5p, and PcDip5) were not conserved in *C. humicola* Aaps (Appendix A) [27].

### 3.2. Growth Characteristics of Chaap4 and Chaap5 Strains on Amino Acids

To investigate the involvement of ChAap4 and ChAap5 in acidic amino acid uptake, we created strains with disruption of each gene (*Chaap4* and *Chaap5*, Figure 1) and analyzed their growth on various amino acids at pH 7 (Table 1). The gene disruptions were not lethal, and they did not result in growth suppression in medium containing glucose as the sole carbon source and NH_4_Cl as the sole nitrogen source, suggesting that the genes might not be necessary for normal cell growth. In addition, the gene-disrupted strains grew on aliphatic l-amino acids, l-methionine, l-serine, and l-dicarboxylic amino acid amides (l-Asn and l-Gln) as the sole nitrogen source, similarly as the wild-type strain. However, *Chaap4* grew more slowly than the wild-type strain on acidic amino acids and l-Phe, and *Chaap5* grew more slowly on l-Lys and l-Phe (Table 1). These results suggested the involvement of ChAap4 in the uptake of acidic amino acids and l-Phe, whereas ChAap5 appears to be involved in the uptake of l-Lys and l-Phe, suggesting that ChAap4 functions as an acidic Aap.

### 3.3. Effect of Medium pH on the Growth of Chaap4 and Chaap5 Strains on Amino Acids

Acidic YATs are suggested to more tightly bind to the unprotonated forms of acidic amino acids [36,37,38], and substrate transport by AdiC and GadC (a Glu/γ-aminobutyrate antiporter of *E. coli*) is pH-dependent [39]. We, therefore, investigated the effect of culture medium pH on the growth of *Chaap4* and *Chaap5* strains on amino acids as the sole nitrogen source (Figure 4). The gene-disrupted strains and wild-type strain displayed similar growth on NH_4_Cl as the sole nitrogen source at all pHs tested (Figure 4A–C). Contrarily, although the *Chaap4* strain grew similarly well as the wild-type strain on l-Asp, d-Asp, and l-Glu at pH 3.0 (Figure 4D,G,J), its growth on l-Asp, d-Asp, and l-Glu was inhibited or completely abolished at pH 10 (Figure 4F,I,L). In addition, on l-Phe, both gene-disrupted strains grew more slowly than the wild-type stain at all pHs tested (Figure 4M,N,O), and the growth of *Chaap4* was more strongly repressed at higher pHs, as observed on acidic amino acids. These results suggested that ChAap4 might function as an acidic Aap and play a critical role in growth on both l-Asp and d-Asp under high alkaline conditions.

As deletion of *ChAAP4* abolished its growth on d-Asp and l-Asp at pH 10 (Figure 4F,I), to clarify the relationship between the function of ChAap4 and ChAap5 and the expression of *ChDDO*, we analyzed DDO activity and *ChDDO* transcription in the *Chaap4* and *Chaap5* strains in the presence or absence of d-Asp as the sole nitrogen and carbon sources at different pHs (Figure 5). Both DDO activity and *ChDDO* transcription were only induced in the presence of d-Asp but not l-Asp and absence of any amino acids. *ChDDO* expression profile was similar between the wild-type and *Chaap5* strains at all pHs tested, and they gradually decreased with increasing pH. Conversely, the induction level of DDO activity and the transcription by d-Asp were markedly lower in the *Chaap4* strain than in the wild-type strain at all pHs, and no activity was detected at pH 10 (Figure 5). These results suggested that the uptake of d-Asp by ChAap4 might participate in the induction of *ChDDO* expression by d-Asp.

## 4. Discussion

Aaps that transport similar amino acids are usually classified into the same clade in a phylogenetic tree (Figure 2) [40]. In Ascomycota fungi such as *S. cerevisiae* (Dip5p) and *A. nidulans* (AgtA), one Aap is usually classified into the acidic Aap clade. Conversely, two Aaps were classified into the acidic Aap clade in *C. humicola* (ChAap4 and ChAap5) and *C. neoformans* (CnAap4 and CnAap5). In this study, one of the two Aaps of *C. humicola*, ChAap4, was suggested to mediate the uptake of acidic amino acids, whereas ChAap5 did not mediate their uptake (Figure 4). Similar to the results of the present study, CnAap4 of an opportunistic pathogen *C. neoformans* is suggested to transport l-Asp, whereas CnAap5 does not transport this amino acid at the growth temperature of 30 °C, although the enzyme is suggested to transport l-Asp at the growth temperature of 37 °C [41]. The two Aaps in each Basidiomycota fungus share high amino acid sequence identity, and the exon-intron structures of their coding genes are identical (Appendix A), suggesting that the genes might have been generated evolutionarily via gene duplication. The gene duplication and the different substrate specificities of the two Aaps might occur to enable the utilization of various amino acids for cell growth. Inconsistency between the substrate specificity and classification in the phylogenetic tree is also found in Aaps of *S. cerevisiae*: The histidine permease Hip1p and tryptophan/tyrosine permease Tat2p are classified into the same clade as the broad substrate specificity permease Gap1p (Figure 2) [17]. As they reported, coclustering in the same clade is not always sufficient to predict the substrate specificity of YATs. It is possible that the existence of multiple Aaps in the same clade indicates that they have different substrate specificities.

The substrate recognition mode in Aaps has been well described in the bacterial l-Arg permease AdiC [35,42]. In the crystal structure of AdiC, the positively charged l-Arg is placed in an occluded chamber with negative surface potential (Figure 6A). This electrostatic interaction is therefore considered to be one contributor to substrate specificity and transport [35]. The electrostatic potential in the substrate-binding site of ChAap4 model structure was suggested to be positive, whereas that of ChAap5 appeared to be more neutral (Figure 6B,C). These findings suggested that the electrostatic potential of ChAap4 might contribute to the substrate preference toward l-Asp and the different substrate preference compared with ChAap5. In addition to the electrostatic interaction, Trp293 in TM8 and Asn101 in TM3 in AdiC are suggested to be involved in the substrate preference by interacting with the guanidinium group of the l-Arg substrate through a π-cation interaction and a hydrogen bond, respectively (Figure 7A) [42]. In ChAap4 model structure, Tyr145 and Ile148 in TM3 were located at the spatially corresponding positions, respectively (Figure 7B). Similar to the amino acid residues of AdiC, these residues might contribute to the substrate preference toward l-Phe of ChAap4 through a π–π interaction and a hydrophobic interaction, respectively. However, it should be noted that in addition to the substrate-binding site, other regions are also known to affect the substrate specificity of Aaps [43]. Further studies, such as mutational studies, are needed to clearly demonstrate the structural factors involved in the substrate specificity of ChAap4.

Fungal Aaps are proton symporters that can transport amino acids using proton gradient (outside > inside cell) as a driving force [19,20]. Therefore, the decreased growth rate on amino acids with increasing pH, observed in this study (Figure 4), might be due to the decreased amino-acid uptake rate with increasing pH, which might cause the significant decrease in the induction of *ChDDO* expression by d-Asp (Figure 5). In fungi, acidic amino acids are transported into cells by both general (nonspecific) and acidic Aaps [36,37,38,43]. It has been reported that the uptake of l-Asp by the general Aap Gap1p decreases with increasing pH [44], and the general amino acid transport system of the fungus *P. chrysogenum* preferentially transports the uncharged forms of l-Glu over the charged forms [37]. On the contrary, the acidic amino acid transporter system of *P. chrysogenum* is suggested to accept both the uncharged and mononegative forms with equal affinities [37]. Together with these findings, the significant growth of *Chaap4* strain (*ChAAP4* gene-disrupted strain) on d- and l-Asp at acidic and neutral pH conditions suggested that unidentified general Aaps as well as ChAap4 might catalyze the uptake of d- and l-Asp at the pH conditions in *C. humicola* (Figure 4D–I), whereas ChAap4 might play a major role in the acidic amino acid uptake at high alkaline conditions. Additionally, the significant growth of the *Chaap4* strain on l-Glu at an alkaline pH implied the presence of a Glu-specific acidic Aap. Alternatively, there is a possibility that a dicarboxylic permease might be involved in l-Glu uptake at higher alkaline pH [45], because approximately 70% of l-Glu is estimated to be present as the dianionic form based on p*K*a = 9.63 of the α-amino group of l-Glu at pH 10. The similar growth phenotype of *Chaap4* strain on d- and l-Asp (Figure 4D–I) and the findings that Gap1p can transport some d-amino acids as well as various l-amino acids [21], d- and l-amino acids are suggested to be transported by a common transport system in *S. cerevisiae* [46], and *Neurospora crassa* possesses an active transporter system for d- and l-acidic amino acids [36] suggesting that ChAap4 could transport both Asp enantiomers.

The transcription of the fungal genes involved in the utilization of amino acids is regulated by various regulatory systems, including nitrogen catabolite repression [47], the SPS sensor system [48], the TOR regulatory pathway [49], and the GAAC pathway [50], by sensing amino acids inside or outside cells. The complicated and sophisticated network creates a broad regulatory system that can regulate the precise control of gene expression. Meanwhile, transcription of *PUT1* and *PUT2*, which encode the proline utilization enzymes proline oxidase and Δ^1^-pyrroline-5-carboxylate dehydrogenase, respectively, is specifically induced only by the presence of l-Pro inside cells. Specifically, l-Pro binds to the transcriptional activator Put3p, which binds to the DNA upstream of the target genes and significantly induces gene transcription [51]. Similarly, as *ChDDO* transcription is specifically induced only in the presence of d-Asp and not subjected to nitrogen catabolite repression [7,8], a d-Asp–specific transcriptional activation system is likely to exist in *C. humicola* strain UJ1. A significant relationship between the growth of *C. humicola* on d-Asp and the transcription of *ChDDO* suggested that d-Asp inside the cells might be recognized by an unknown d-Asp–specific sensor protein or transcriptional activator protein that can induce the transcription.

## 5. Conclusions

In conclusion, we propose the following model for the relationship between the inducible expression of *ChDDO* in the presence of d-Asp and the uptake of d-Asp by ChAap4 and other Aaps in *C. humicola* strain UJ1 (Figure 8). In acidic and neutral environments, d-Asp uptake might be mediated by multiple Aaps, including ChAap4 and Gap(s). The d-Asp uptake activity of Gap(s) might be gradually decreased by changing the environmental pH from neutral to alkaline conditions and completely abolished under highly alkaline conditions, whereas ChAap4 could still import d-Asp even under high alkalinity. d-Asp imported by the Aaps might induce the transcription of *ChDDO* to utilize d-Asp for cell growth and reduce d-Asp-mediated toxicity. At present, the sensor that recognizes d-Asp inside cells and induces gene transcription is unknown, and future research should aim to identify the involved protein(s) to clarify the entire mechanism of the d-Asp–specific gene transcriptional activation. The clarification of the whole regulation mechanism of *DDO* gene expression in the yeast might contribute to the further understanding of the physiological functions of DDO and d-Asp in not only fungi but also animals including human, and, furthermore, the onset mechanism of the psychiatric disease schizophrenia suggested to be associated with DDO and d-Asp.

## Figures and Tables

**Figure 1 microorganisms-09-00192-f001:**
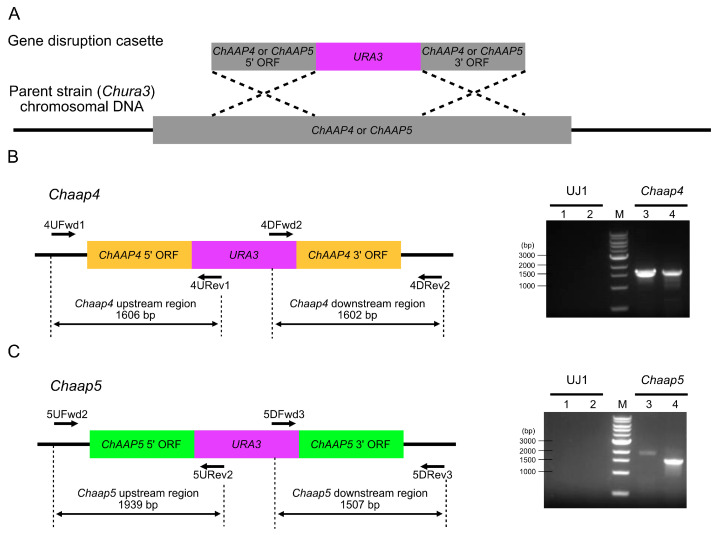
Construction of Chaap4 and Chaap5 strains. (**A**) Schematic representation of *ChAAP4* and *ChAAP5* gene disruptions by homologous recombination using *ChURA3*. (**B**) PCR analysis of *ChAAP4* gene disruption. The numbers above the box indicate the distance (bp) from the start codon ATG of the chromosomal *ChAAP4*. *ChURA3* was inserted, in the opposite direction, into the *ChAAP4* between 117 and 309 bp. Lanes 1 and 2, the negative control amplification of the upstream and the downstream regions, respectively, in the wild-type strain (strain UJ1); lanes 3 and 4, the amplification of the upstream and downstream regions in *Chaap4* strain. (**C**) PCR analysis of *ChAAP5* gene disruption. The numbers above the box indicate the distance (bp) from the start codon ATG of the chromosomal *ChAAP5*. *ChURA3* was inserted, in the opposite direction, into the genomic *ChAAP5* between 716 and 1368 bp. Lanes 1 and 2, the negative control amplification of the upstream and the downstream regions, respectively, in the wild-type strain (strain UJ1); lanes 3 and 4, the amplification of the upstream and the downstream regions in *Chaap5* strain. PCR was performed using *Chaap4-* or *Chaap5*-specific primer pairs in Appendix A.

**Figure 2 microorganisms-09-00192-f002:**
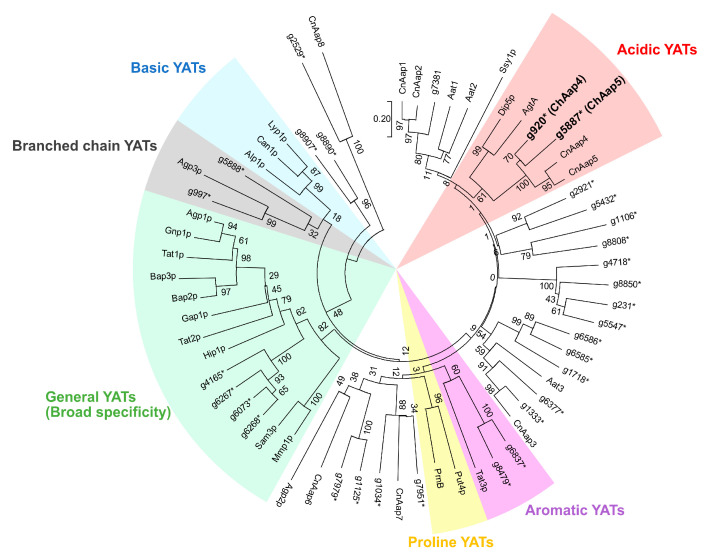
Phylogenetic relationship of Aaps homologs of *C. humicola* strain UJ1 and YATs. The phylogenetic tree was constructed by the neighbor-joining method with 1000 bootstrap trials using MEGA version 7.0. The numbers at nodes indicate bootstrap value as percentages. The asterisks indicate Aap homologs of strain UJ1. Accession numbers of the amino acid sequences used for the analysis were as follows: *C. neoformans* CnAap1 (CNAG_02539), CnAap2 (CNAG_07902), CnAap3 (CNAG_1118), CnAap4 (CNAG_00597), CnAap5 (CNAG_07367), CnAap6 (CNAG_07449), CnAap7 (CNAG_05345), and CnAap8 (CNAG_00574); *Saccharomyces cerevisiae* Tat1p (Uniprot: P38085), Tat2p (P38967), Tat3p (A4UZ28), Gap1p (P19145), Hip1p (P06775), Gnp1p (P48813), Agp1p (P25376), Agp2p (P38090), Agp3p (P43548), Bap2p (P38084), Bap3p (P41815), Sam3p (Q08986), Mmp1p (Q12372), Lyp1p (P32487), Alp1p (P38971), Can1p (P04817), Dip5p (P53388), Put4p (P15380), and Ssy1p (Q03770); *Aspergillus nidulans* AgtA (B2M1L6) and PrnB (P18696); *Uromyces fabae* Aat1 (Q96TU9), Aat2 (O00062), and Aat3 (Q700T6).

**Figure 3 microorganisms-09-00192-f003:**
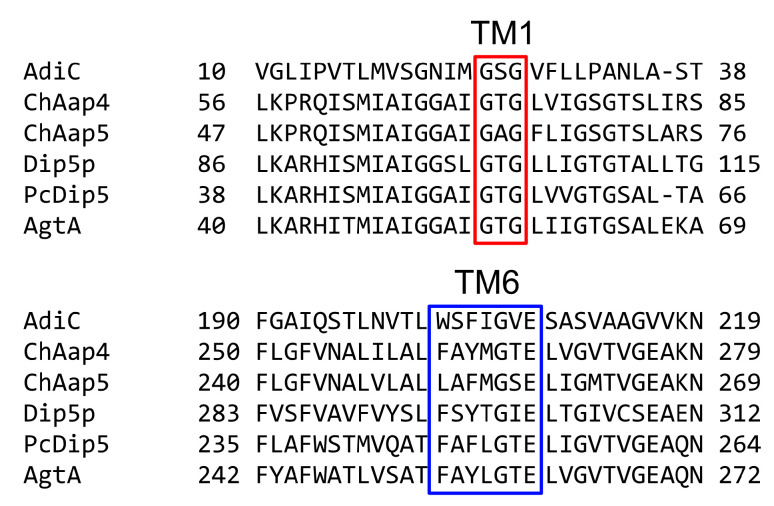
Comparison of amino acid sequences at transmembrane (TM) region 1 and 6 of ChAap4 and ChAap5 with *E. coli* AdiC (UniProKB: P60061) and three yeast dicarboxylic amino acid permeases: Dip5p (*S. cerevisiae*), PcDip5 (*P. chrysogenum*), and AgtA (*A. nidulans*). Transmembrane (TM) regions were predicted by Phobius (http://phobius.sbc.su.se/). GXG and (F/Y)(S/A/T)(F/Y)XGXE motifs (where X is any amino acid) are boxed in red and blue, respectively.

**Figure 4 microorganisms-09-00192-f004:**
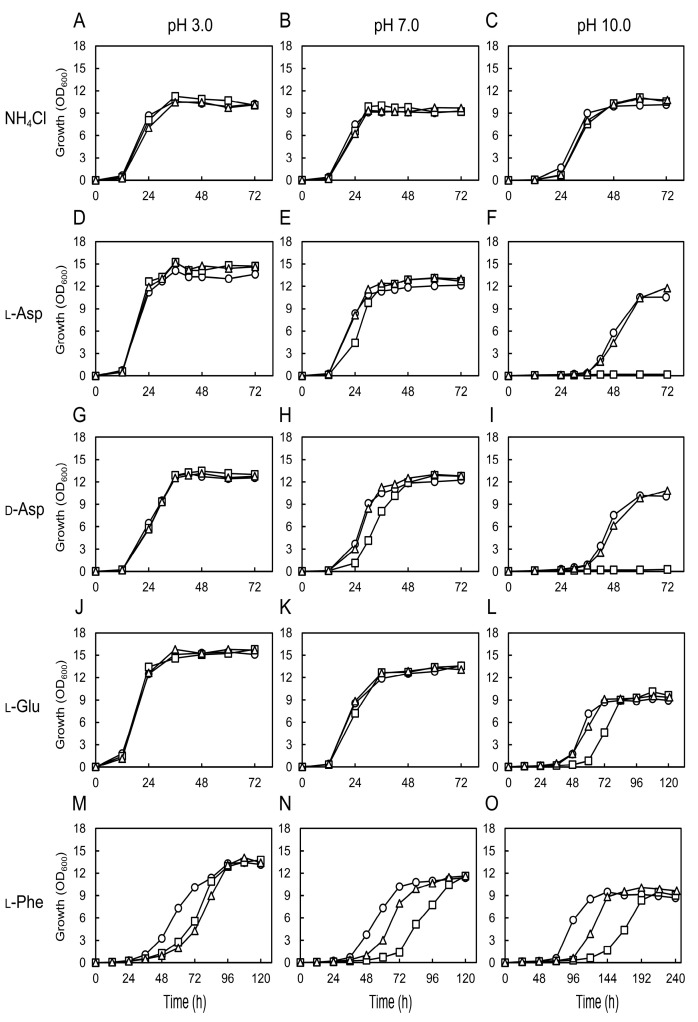
Effect of medium pH on the growth of *Chaap* strains on amino acids. The wild-type (strain UJ1) (open circles), *Chaap4* (open squares), and *Chaap5* (open triangles) strains were cultivated at 30 °C in a medium containing 10 mM NH_4_Cl or each amino acid as the sole nitrogen source. Initial pH of the media was adjusted to 3.0 (**A**,**D**,**G**,**J**,**M**), 7.0 (**B**,**E**,**H**,**K**,**N**), or 10.0 (**C**,**F**,**I**,**L**,**O**).

**Figure 5 microorganisms-09-00192-f005:**
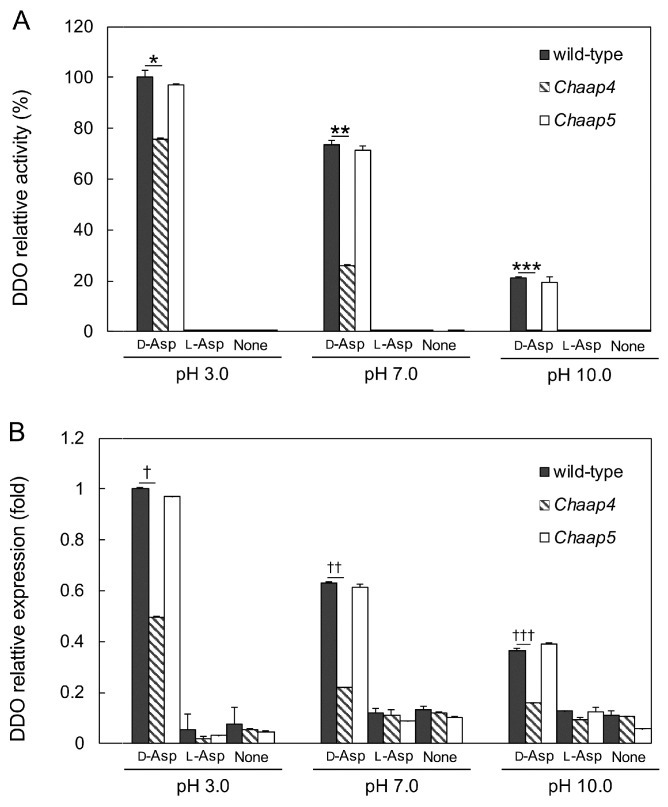
The expression of *ChDDO* gene in *Chaap* strains grown on d-Asp at different pHs. (**A**) DDO activity in the extracts from the cells grown in a synthetic medium containing (d-Asp or l-Asp) or not containing (None). For this, 60 mM d-Asp or l-Asp was used as the sole nitrogen and carbon sources at different pHs at 30 °C. The enzyme activity is expressed as a percentage of that of the wild-type strain at pH 3.0. Statistical differences were ascertained by Student’s *t*-test, * *p* < 5 × 10^−4^, ** *p* < 5 × 10^−6^, and *** *p* < 1 × 10^−6^. (**B**) Transcription of *ChDDO* gene. The gene transcription was analyzed by qRT-PCR using total RNA from cells grown under the same conditions as in the analysis of DDO activity, normalized to *TAF10* gene transcription, and expressed as a relative ratio of that of the wild-type strain at pH 3.0. Statistical differences were ascertained by Welch’s *t*-test, † *p* < 5 × 10^−4^, †† *p* < 5 × 10^−7^, and ††† *p* < 1×10^−7^. The values are the means of three independent experiments, and the error bars are the standard deviations.

**Figure 6 microorganisms-09-00192-f006:**
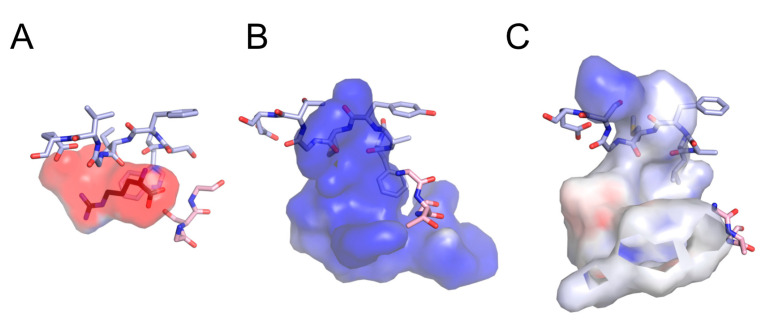
Electrostatic potential of substrate-binding pockets of AdiC (**A**), ChAap4 (**B**), and ChAap5 (**C**). The electrostatic potential was calculated using the software PyMOL 2.3x and is showed by a gradient of blue (positive charge) and red (negative charge) colors. The carbon atoms of l-Arg in AdiC is colored in black. Oxygen and nitrogen atoms are shown in red and blue, respectively. GXG and (F/Y)(S/A/T)(F/Y)XGXE motifs are displayed in light pink and light blue, respectively.

**Figure 7 microorganisms-09-00192-f007:**
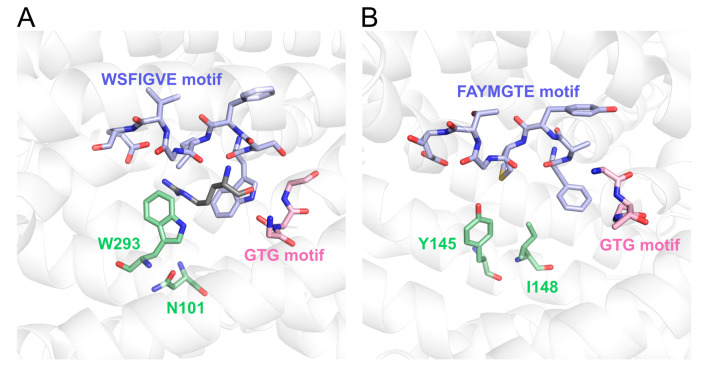
Substrate-binding site of AdiC (**A**) and ChAap4 (**B**). Trp293 in TM8 and Asn101 in TM3 in AdiC and Tyr145 and Ile148 in TM3 in ChAap4 potentially involved in the substrate preference are displayed in light green.

**Figure 8 microorganisms-09-00192-f008:**
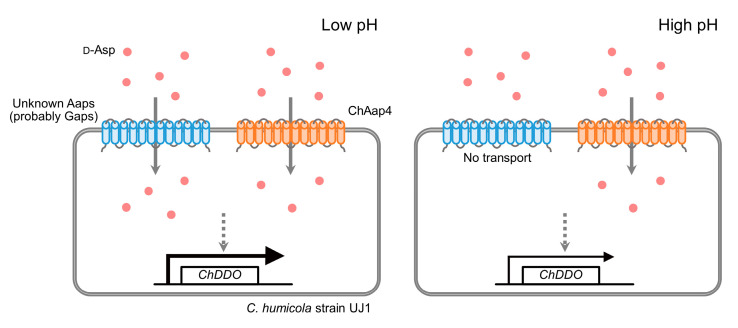
Relationship between d-Asp uptake by ChAap4 and *ChDDO* gene expression in the yeast *C. humicola* strain UJ1. In acidic and neutral environments, d-Asp molecules (red circles) are transported (grey arrows) by ChAap4 (red proteins) and unknown Aaps (probably Gaps, blue proteins). In high alkaline environments, the d-Asp uptake by Gaps is abolished but not by ChAap4. Intracellular d-Asp induces *ChDDO* gene expression via an unknown signaling pathway (grey dotted arrows).

**Table 1 microorganisms-09-00192-t001:** Growth of *Chaap4* and *Chaap5* strains on various amino acids.

	N Source	*Chaap4*	*Chaap5*
	None	Below Detection	Below Detection
	NH_4_Cl	95%	98%
Aliphatic	l-Ala	106%	115%
	Gly	108%	115%
	l-Ile	95%	107%
	l-Val	113%	104%
Aromatic	l-Phe	60% *	79% *
Sulfur	l-Met	102%	95%
Hydroxylated	l-Ser	106%	106%
Acidic	l-Asp	36% ***	94%
	d-Asp	67% **	102%
	l-Glu	84% **	103%
Amide	l-Asn	96%	107%
	l-Gln	102%	108%
Basic	l-Lys	96%	92% *

The gene-disrupted strains were cultivated at 30 °C at pH 7.0 in a synthetic media containing 2% (*w/v*) glucose and 10 mM each amino acid as the sole nitrogen source and their growth at OD_600_ are expressed as percentages of that of the wild-type strain in mid-logarithmic phase. Statistical differences were ascertained by Student’s *t*-test, * *p* < 0.05, ** *p* < 0.01, and *** *p* < 0.001 (*n* = 3 for each strain).

## Data Availability

The data presented in this study are available on request from the corresponding author.

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
