# Peer review of "Identification of an Acidic Amino Acid Permease Involved in d-Aspartate Uptake in the Yeast Cryptococcus humicola"

_microorganisms, 2021, doi:10.3390/microorganisms9010192_

Round 1

Reviewer 1 Report

The title is misleading: the authors demonstrated that d-asp in internalized thorught Chaap4 and, probably, by another not yet identified system. It is the d-asp concentration that is involved into the expression of DDO.

check some English errors: L26 peroxisomal; L29:add a full stop; All manuscript: D is in caps in words like wilD-type and it is small caps in d-Asp; L205: TM regions; L314:Dicarboxylic acid anion: replace with dianionic

Introduction: L35-41 there is a lot of information concerning the role of D-Aps and DDO from microorganism to mammals. I think this review is not necessary in this context. Please reduce and focus more on role of amino acids as nitrogen/carbon source for microorganisms.

Although quite long, it is not clear the importance of the performed work (from a basic and applied research)

L86: ChAap4 might….. and the inducible …. as stated above, the last part of the sentence is misleading since ChAap4 is not directly involved into the expression of DDO. It is involved in the uptake of D-Asp.

Tab 1: n.g. should be replaced by Below detection and n.s. should be replaced by numeric data (e.g. 99%, 100%, 1%....)

A major discussion point emerges (see also fig.3): the effect of deletion of ChAap4 on growth in L-Phe.
I think that the authors should elaborate more this point. From a structural point of view, it is unclear that a transporter is specific for L-Phe and L-Asp which possess different shape, size, polarity. The authors should test for example Tyr and terephthalic acid (or similar compounds).

General consideration: I suppose that this kind of analysis (i.e. the relationship between a compound and the growth rate) suffer from the fact that the observed effect (i.e. the growth o strain) depends on several processes (intake ability, ability to use the compound, potential toxicity, …).

Fig. 3: a more detailed profile of growth vs pH for the WT and at least NH4Cl and D-Asp should be interesting (from 2 to 12), just to evaluate for the reader the effect of pH on growth. From your data seems that there is NO effect of pH in the range 3-10.

I think that pH10 is too low in comparison with the pKa (9.5-9.8) of the amino group of the amino acids. At this pH there is a mixture of the protonated and not protonated form, so it is not useful for understanding the effect of the protonation of the amino group of the amino acid and its transport. I suggest to repeat panel C,F,I,L,O at pH 11 (or at least 10.5) where almost all amino acid molecules are deprotonated.

Fig.4: There is a discrepancy with fig. 3. The WT growth on D-Asp is unaffected by the the pH (panel I) but the DDo expression is significantly lower? Can this be explained?
At pH10 in Chaap4 strain, still some expression is observed. Why not activity?

A same analysis should be done with L-Asp as a control.

Reviewer 2 Report

Imanishi with colleagues presented an interesting study of amino acid permease enzyme identification involved in d-asparated oxidase gene expression in C. humicola. In my opinion the study was carried out with great detail and deserves publishing after small modifications. Figure 3 is hard to follow - maybe please move the C/N source to the left of the panels and make them larger, same with pH values on the top. Please move Figure S2 to the main body of the article as it will be beneficial to the readers. Science is great but I am lacking a practical sum up of your work - please include few sentences drawing a bigger picture in the conclusion section. 

Author Response

Please see the atachment.

Round 2

Reviewer 1 Report

The authors positively answered to my concerns.